# Monomeric and Oligomeric Decorsins of the Asian Medicinal Leech *Hirudinaria manillensis*

**DOI:** 10.3390/ijms262211017

**Published:** 2025-11-14

**Authors:** Céline Tolksdorf, Robert Wolf, Bernhard H. Rauch, Gabriele Jedlitschky, Christian Müller

**Affiliations:** 1Pharmacology and Toxicology, University Medicine Oldenburg, Carl von Ossietzky University Oldenburg, 26129 Oldenburg, Germany; celine.tolksdorf@uni-oldenburg.de (C.T.); bernhard.rauch@uni-oldenburg.de (B.H.R.); 2Section of Nephrology, Clinic and Policlinic of Internal Medicine A, University Medicine Greifswald, 17489 Greifswald, Germany; robert.wolf@med.uni-greifswald.de; 3Department of General Pharmacology, University Medicine Greifswald, 17489 Greifswald, Germany; gabriele.jedlitschky@med.uni-greifswald.de; 4Animal Physiology, Zoological Institute and Museum, University of Greifswald, 17489 Greifswald, Germany

**Keywords:** leech, *Hirudo manillensis*, anticoagulants, decorsin, platelet aggregation

## Abstract

Hematophagous leeches rely on a broad diversity of bioactive factors that interfere with the host’s defense systems to secure a successful blood meal. The most prominent examples of such factors are the antithrombotics that address either the primary hemostasis (the platelet aggregation) or the secondary hemostasis (the blood coagulation). Whereas the inhibitors of platelet aggregation mainly cause continuous blood flow, coagulation inhibitors mainly keep the blood fluid within the stomach of the leech. The critical dependency of hematophagous leeches on the accurate action of all antithrombotic factors may explain the presence of multiple genes for each type of these factors that is regularly observed in leech genomes. The genome of *Hirudinaria manillensis* Lesson, 1842, the Asian buffalo leech, contains five individual genes that encode variants of the coagulation inhibitor hirudin. However, no genes that encode putative decorsins, the archetype of leech-derived inhibitors of platelet aggregation, have been described to date. Here, we report the identification of one monomeric and four multimeric decorsin encoding genes in the genome of *H. manillensis*. A selection of the putative decorsins was expressed as recombinant proteins, purified and functionally characterized. All but one of these putative decorsins displayed platelet aggregation-inhibitory potencies. Furthermore, we provide a mathematical calculation based on the mechanism of alternative pre-mRNA splicing that illustrates the potential to generate an enormous variety of different factors from one single multimeric ornatin gene.

## 1. Introduction

Leeches of the genus *Hirudinaria* Whitman, 1886, also known as buffalo leeches, belong to the family of Hirudinidae [1]. They are common to South-East Asia [2] but were also introduced to the West Indies [3]. They inhabit freshwater environments including swamps and rice fields and mainly feed on larger mammals like cattle but also attack humans [4]. To date the genus *Hirudinaria* comprises four different species, namely *Hirudinaria manillensis* Lesson, 1842, *Hirudinaria javanica* Wahlberg, 1856, *Hirudinaria bpling* Phillips, 2012, and *Hirudinaria thailandica* Jeratthitikul & Panha, 2020 [5,6]. The different species may occur sympatrically [7]. Their saliva comprises antimicrobial and anticoagulant activities [8], and they are used for medical purposes such as the treatment of venous congestion in microvascular surgery [9] or wound healing and pain management [10]. Consequently, *H. manillensis* is also known as the “Asian medicinal leech” [11]. The leeches are either still collected from the wild or bred in farms and sold as living individuals to practitioners or dried and exported in large quantities to China [12].

Already in the early 1990s, the first hirudins were described in *H. manillensis* [13,14,15]. Later, several other putative antithrombotic factors were identified, including Kazal-type inhibitors (bdellin-HM, [16]; bdellin-HM2, [17]) and hirudin-like factors (HLFs) [18,19,20,21]. Strikingly, one of the HLFs comprised a duplication of the central globular domains and was hence termed tandem-hirudin (TH) [20]. The central globular domain is a structural hallmark of all hirudins and HLFs, as well as decorsins [22] and ornatins [23] (both factors are inhibitors of platelet aggregation), and defines the so-called hirudin superfamily [24]. The domain contains six conserved cysteine residues, and its structure is stabilized by the formation of three disulfide bonds [25]. On the gene level, the domain is encoded by two exons, namely the exons 2 and 3, of the complete hirudin gene that in total comprises four exons [15,26]. The structure of the respective exons 2 and 3 allows for a duplication (forming tandem structures like TH) or even an oligomerization (forming several repeats of the globular domain). The oligomerization hypothesis was developed by Lukas et al. [20] and recently confirmed by the identification of a putative oligomeric decorsin of the Indian land leech *Haemadipsa interrupta* Moore, 1935, which contains three repeats of the globular domain [24]. The idea that exons encode functional units that can be arranged to form new proteins has been a topic of debate for decades [27,28], and the globular domain of hirudins can be defined as such a functional unit, albeit spread over two exons.

Dimerization and/or oligomerization tendencies have also been described for antistasin-like proteins [29], the leech-derived tryptase inhibitor (LDTI) [30,31], and the platelet aggregation inhibitor saratin [32,33,34]. Probably the most impressive example of oligomerization, however, is progranulin, a growth factor-like, neurotrophic and immunomodulatory protein [35] that is composed of tandem repeats of the granulin/epithelin module [36]. Progranulin is processed by neutrophil proteases to release the individual granulin modules. Leech-derived progranulins contain between 8 and 14 granulin modules [34]. Strikingly, the granulin of *Hirudo nipponia* Whitman, 1886, exhibits thrombin-inhibitory potency [37]. However, there is currently no evidence that proteolysis is involved in the processing of oligomeric leech-derived antithrombotic factors apart from progranulin.

Alternative pre-mRNA splicing is a mechanism that allows a single gene to produce different mRNA splice variants and hence different proteins with potentially different functions [38]. So far, alternative pre-mRNA splicing of leech-derived putative antithrombotic factors has been described for HLF3 of *H. medicinalis* [26], the hirudin variant Wpig_HV1 of *Whitmania pigra* Blanchard, 1887 [39], and antistasin-like proteins of *Macrobdella decora* Say, 1824 [40]. An in-depth analysis of the occurrence and frequency of alternative pre-mRNA splicing in leeches based on a combination of transcriptome and genome data is still missing. However, an integration of alternative pre-mRNA splicing into the oligomerization hypothesis of the globular domain of hirudins offers almost infinite possibilities to boost the number of different splice variants and hence the diversity of putative anticoagulant proteins.

Starting with the genome of *Helobdella robusta* Shankland, Bissen & Weisblat, 1992 [41], recent advances in whole genome sequencing technologies led to a rapid increase in both the number and the quality of fully sequenced draft [42,43,44,45] and chromosome-level [31,34,46,47,48,49,50] leech genomes. Data analyses of leech genomes mainly focus on the identification of genes that encode putative antithrombotic factors. The number of annotated different genes can rise up to more than 120 [34], a value that is in good proximity to estimations of leech saliva components that were obtained with completely different technological approaches based on 2D-gel electrophoresis [51] or MALDI-TOF MS [52]. Gene annotation attempts critically depend on the availability of suitable query sequences and the appropriate use of bioinformatic tools [53]. Consequently, thorough re-investigations of genome data may lead to the identification of yet missed genes. Recently, full genome data of *H. manillensis* were independently provided by Guan et al. [44], Liu et al. [31] and Zheng et al. [46], and in all three genomes concordantly, five genes that encode putative hirudins and HLFs could be identified [20,31]. However, no study has yet reported the identification of putative decorsins. The aim of the present study was to re-analyze all available genome data of *H. manillensis* for the presence of decorsins or decorsin-like factors and, if successful, to express and functionally characterize the recombinant proteins.

## 2. Results

### 2.1. Identification of Putative Decorsin Genes

Three different genome data sets of *H. manillensis* are available in GenBank, generated by Guan et al. [44], Zheng et al. [46], and Liu et al. [31]. In all three genomes, five individual genes encoding hirudins, hirudin-likes factors (HLFs), and a tandem-hirudin (TH) can be identified, located on contig00065 [20,44], chromosome 3 [46], or chromosome 4 [31], respectively.

To date, no genes that encode putative decorsins have been described. We performed an in-depth re-analysis of all three genomes and the two transcriptome data sets, using the amino acid sequences of different decorsins including the decorsins of *L. mexicana* and *H. vizottoi* [54] as queries. In total, we were able to identify five individual genes in each of the three genomes that encode putative decorsin variants (DV), namely Hman_DV1-5. All putative decorsin genes are located on the same genetic entities as the corresponding hirudin/HLF/TH genes (contig00065, chromosome 3 or chromosome 4, respectively), but are orientated into the opposite direction and separated from the hirudin/HLF/TH genes by a distance of about 120–140 kb. The exact positions of all hirudin/HLF/TH genes and the putative decorsin genes in all three genomes is provided in Table 1, and a multiple sequence alignment (MSA) of the complete putative decorsin gene ranges including an annotation of the genes is provided in Appendix A.

Whereas Hman_DV4 is a putative monomeric decorsin, the gene of Hman_DV2 encodes a putative tetrameric decorsin, and Hman_DV1, _DV3 and DV5 are putative hexameric decorsins. All putative decorsin genes comprise the typical structure of genes that belong to the hirudin superfamily: the first exon encodes the signal peptide, whereas the following exons encode a single (Hman_DV4) or multiple repeats (Hman_DV1, 2, 3 and 5) of the globular domain. The genes of Hman_DV4 (monomer), Hman_DV2 (tetramer), and Hman_DV1 (hexamer) are hence composed of 3, 9 or 13 exons, respectively. However, the genes of Hman_DV3 and Hman_DV5 additionally encode a short C-terminal tail and hence comprise 14 exons. The only exception is the putative decorsin gene Hman_DV5 in the genome data set of Liu et al. [31] that lacks the exons 2–4, presumably due to a large deletion. Figure 1 shows a schematic representation of size, localization, and orientation of all putative decorsin genes and the detailed structures of the genes for Hman_DV1 and Hman_DV4 based on contig00065. MSAs of related putative decorsin gene sequences (e.g., the Hman_DV1 genes of all three genomic resources) including detailed annotations of exons and introns are provided in Appendix A.

A hallmark of all platelet aggregation-inhibitors of the decorsin/ornatin-type is the presence of an RGD (or KGD) motif that is located between the fifth and the sixth cysteine residues [55,56]. In all five putative decorsins of *H. manillensis*, respective molecular signatures could be identified. Concordantly, the motifs are located within the first repeat of the globular domain, but the putative decorsins Hman_DV2 and Hman_DV3 each contain a second motif either in globular domain repeat 4 (Hman_DV2) or globular domain repeat 3 (Hman_DV3, see Figure 1). Table 2 summarizes structural and biochemical features of all five putative decorsins of *H. manillensis*. MSAs of the respective putative decorsin protein sequences including labelling of the cysteine residues and the RGD/KGD motifs are provided in Appendix A.

The globular domains in multimeric decorsins that contain the RGD/KGD motifs can be part of the whole multimeric protein, but they can potentially also be expressed as individual monomers due to alternative splicing events. Two of the main mechanisms of alternative splicing are intron retain and exon skipping [38]. To illustrate examples for Hman_DV1 and Hman_DV5, if intron I3 in both pre-mRNA2 is retained, the first stop codon within intron I3 in frame with exon 2 will terminate the translation of the respective mRNA and a monomeric factor will be formed. The putative monomeric decorsins were designated Hman_DV1s and Hman_DV5sa (“s” indicates “short”, see Figure 2 and Table 3). In addition, the exons E4-E13 of the Hman_DV5 gene can be removed by exon skipping, and hence, exon E3 can be directly spliced with exon E14 (exon E14 encodes a short C-terminal tail, see Appendix A), giving rise to the putative monomeric decorsin Hman_DV5sb.

Both the Hman_DV2 and Hman_DV3 genes contain two exons that encode RGD/KGD motifs, namely exons E3 and E9 in Hman_DV2 and exons E3 and E7 in Hman_DV3 (see Figure 1 and Appendix A). If intron I3 is retained in both pre-mRNAs, the monomeric putative decorsins Hman_DV2s-1 and Hman_DV3s-1a will be formed. However, exon E3 of the Hman_DV3 gene can alternatively also be spliced with exon E14 (exon E14 encodes a short C-terminal, see Figure 1 and Appendix A), and the deduced putative monomeric decorsin was termed Hman_DV3s-1b. The second RGD/KGD motif encoding exons in both genes (exon E9 in Hman_DV2 and exon E7 in Hman_DV3, see Figure 1) can alternatively be spliced with all even upstream exons, namely exons E2, E4, E6, and E8 in Hman_DV2 and exons E2, E4, and E6 in Hman_DV3, respectively. If the adjacent downstream introns (I9 of the Hman_DV2 gene and I7 of the Hman_DV3) are retained, the gene of Hman_DV3 potentially encodes four and the gene of Hman_DV2 three additional putative monomeric decorsins (designated Hman_DV2s-2-5 and Hman_DV3s-2a-4a). Alternatively, the RGD motif-containing exon E7 of Hman_DV3 can also be spliced with exon E14, the respective factors were termed Hman_DV3s-2b-4b. Taken together, we have deduced 16 putative monomeric decorsins from the four putative multimeric decorsin genes. Figure 2 provides an MSA of all deduced putative monomeric decorsins of *H. manillensis*, and Table 3 summarizes the biochemical features of the respective factors.

The putative monomeric decorsins of *H. manillensis* considerably differ in their molecular masses (between about 4.5 and 7.1 kDa) and cover a broad range of isoelectric points (between basic 4.08 and acidic 9.33).

### 2.2. Calculation of Putative Decorsin Diversity

As already mentioned, two key mechanisms of alternative splicing are exon skipping and intron retain [38]. Both mechanisms can be applied to characterize the alternative splicing events that form repeats of the globular domain in putative decorsins of *H. manillensis* and to calculate the number of different proteins that are encoded by a single putative decorsin gene. Figure 3 illustrates the mechanism of exon skipping using a partial Hman_DV3 gene as a template. Due to the molecular characteristics of the splicing sites (exon-overlapping or non-overlapping codons), uneven exons can only be spliced with even exons and even exons can only be spliced with uneven exons to form meaningful open reading frames (ORFs).

Taken together, the Hman_DV3 gene comprises the capability to encode a total of 196 different factors that contain one or more repeats of the globular domain: 1 hexameric factor, 11 pentameric factors, 32 tetrameric factors, 61 trimeric factors, 70 dimeric factors, and 21 monomeric factors. Two RGD/KGD motifs are present in 27 factors, and 99 factors contain at least one of the motifs, forming a total of 126 different putative decorsins. Appendix A lists all possible combinations of globular domains that can be deduced from the Hman_DV3 gene by regular and alternative splicing using the exon skipping mechanism. If intron retain is considered an additional mechanism of alternative splicing, the numbers double—all uneven introns contain stop codons that are in frame with uneven exons.

### 2.3. Functional Characterization of Putative Decorsins

A representative selection of putative monomeric decorsins, namely Hman_DV1s, Hman_DV3s-1a, Hman_DV3s-4a, and Hman_DV4 (see Figure 2 and Table 3) in addition to the multimeric decorsin Hman_DV1, was made for recombinant expression, purification, and functional investigation of the respective factors. The factors were selected among other criteria based on the transcriptomic data provided by Guan et al. [44] and Zheng et al. [46]. All selected factors were successfully expressed, purified (gel images to illustrate the His-tag purification steps of all factors are provided as Appendix A), and functionally characterized for their platelet aggregation-inhibitory potencies. For technical and logistical reasons, the respective experiments were performed at two different locations and with two different aggregometers (see Section 4 for details). As can be seen in Figure 4, all monomeric factors displayed clear platelet aggregation-inhibitory potencies and can hence be termed “decorsins”. The multimeric factor Hman_DV1, however, did not negatively affect platelet aggregation.

## 3. Discussion

The Asian medicinal leech, *H. manillensis*, encodes five different hirudins and hirudin-like factors. Among them was the first oligomeric representative of the hirudin superfamily ever found, a tandem-hirudin (TH; [20]). The TH, however, did not inhibit thrombin. Only recently a first oligomeric decorsin (Hint_DV1) was identified in *H. interrupta*, the Indian land leech. Recombinant Hint_DV1 displayed a low, but clearly detectable potency to inhibit platelet aggregation [24]. Putative decorsins have not been described in *H. manillensis* to date. In the present study, we were able to identify five genes that encode putative decorsins in all three available *H. manillensis* genomes, namely Hman_DV1-5, using different decorsin variants of different leech species as query sequences. Strikingly, four genes encode putative oligomeric decorsins that contain four (Hman_DV2) or six (Hman_DV1, DV3, and DV5) repeats of the globular domain. Only one gene (Hman_DV4) encodes an archetype monomeric factor. All factors contain at least one RGD or KGD motif, a hallmark of all leech-derived platelet aggregation inhibitors of the decorsin/ornatin-type [55,56]. Notably, two of the hexameric factors comprise two RGD/KGD motifs—a unique feature that has not been described before. The putative hexameric decorsin Hman_DV1, the putative monomeric decorsin Hman_HV4, and additionally three putative monomeric decorsins (Hman_DV1s, Hman_DV3s-1a, and Hman_DV3s-4a, see Figure 2 and Table 3) that were deduced from the respective hexameric genes were selected for recombinant expression in *E. coli* DH5α. The recombinant expression of cysteine-rich proteins in bacterial systems is a challenging task for at least two reasons:(1)The proteins are often found in inclusion bodies [57], a problem that was also observed for the tandem-hirudin of *H. manillensis* [20];(2)The correct formation of disulfide bonds is uncertain [58,59].

In the studies described here, none of the recombinant putative decorsins of *H. manillensis* were present in inclusion bodies, and all factors could be successfully expressed, purified, and functionally tested in platelet aggregation assays. All monomeric factors displayed the ability to interfere with ADP-induced platelet aggregation (see Figure 4), demonstrating the potential ability of *H. manillensis* to modulate platelet function in its vertebrate hosts. The putative hexameric decorsin Hman_DV1, however, did not affect platelet aggregation, whereas the derived factor Hman_DV1s did (see Figure 4). It is likely that the recombinant Hman_DV1 was not correctly folded. The factor contains 36 cysteine residues and hence requires the correct formation of 18 disulfide bonds. The critical RGD/KGD motif in decorsins is located at the apex of a dynamic loop [60], and incorrect disulfide bond formation may disturb or even prevent the forming of the respective loop. Hint_DV1 of *H. interrupta* is a trimeric decorsin that requires the correct formation of only nine disulfide bonds. The factor exhibited a comparably low, yet detectable platelet aggregation-inhibitory potency [24], indicating that only a fraction of the recombinant Hint_DV1 was correctly folded.

Besides inhibition of platelet aggregation, RGD-containing peptides and toxins may exhibit additional physiological effects [61]. Lj-RGD3 is a three-RGD-motif-containing protein from the salivary gland secretions of *Lampetra japonica* Martens, 1868, the Arctic lamprey that, comparable to hematophagous leeches, feeds on blood and other body fluids of its hosts [62]. Recombinant Lj-RGD3 displays neuroprotective effects [63]. It remains to be proven whether or not the multimeric decorsins of *H. manillensis* have similar or additional effects, but they are certainly promising candidates for future drug developments.

However, the diversity of putative bioactive factors that are encoded by the multimeric decorsin genes of *H. manillensis* is even higher. Alternative splicing in combination with the oligomerization of the globular domain may lead to the synthesis of hundreds of different factors from a single gene only. As outlined above, the Hman_DV3 gene comprises the potential to express almost 200 different factors, ranging from monomers to the hexameric DV3 protein (see Appendix A for a complete list), solely based on the mechanism of exon skipping. If intron retain is considered an additional mechanism, the number doubles. It can hence be roughly estimated that the four oligomeric decorsin genes of *H. manillensis* together comprise the potential to express more than 1000 different factors of the hirudin superfamily, and each of these factors may possess specific biological functions and putative therapeutic benefits. Medicinal leeches can truly be considered living drug stores [64].

## 4. Materials and Methods

### 4.1. Genome and Transcriptome Data

Genome and transcriptome data of *H. manillensis* were obtained from the following sources:

Genome generated by Guan et al. [44]:

GenBank assembly GCA_015345955.1;

GenBank SRA database (accession number SRR7415780).

Transcriptome data generated by Guan et al. [44]:

GenBank SRA database (accession numbers SRX6035209–SRX6035214).

Genome and transcriptome data generated by Zheng et al. [46]:

NCBI BioProject PRJNA762643;

Figshare online repository: https://doi.org/10.6084/m9.figshare.20400729.

Genome data generated by Liu et al. [31]:

GenBank assembly GCA_034509925.1;

Figshare online repository: https://doi.org/10.6084/m9.figshare.24187377.v1.

### 4.2. Bioinformatics and Graphical Tools

Basic Local Alignment Search Tool (BLAST) searches were performed using either the NCBI web portal https://blast.ncbi.nlm.nih.gov/Blast.cgi (accessed on 18 August 2025) or BioEdit v7.2.5 [65] and adjusted parameters for both word size and the expected threshold values. Multiple sequence alignment (MSA) files were generated using either ClustalX 2.1 [66] or the CLC Sequence Viewer software package v8.0 (QIAGEN, Aarhus, Denmark) with default settings. Respective alignments were exported as msf-files and further processed using Gene-Doc v2.7 [67]. Signal peptide sequences were predicted using SignalP6.0 [68]. Graphs were generated and analyzed using GraphPad Prism V5.01 (GraphPad Software, Boston, MA, USA).

### 4.3. Gene Synthesis

cDNA fragments of putative hirudins and HLFs were generated using the GeneArt gene synthesis service of ThermoFisher Scientific (Darmstadt, Germany).

### 4.4. Amplification and Cloning of Putative Monomeric and Oligomeric Decorsin cDNAs

For the amplification of putative monomeric and oligomeric decorsin cDNAs, primers were derived from the respective genome sequences. A list of all primers that were used in the study is provided in Appendix A. PCR reactions were performed in a final volume of 25 µL using Q5 high-fidelity DNA polymerase (New England Biolabs, Frankfurt am Main, Germany) and the following protocol: 3 min 98 °C initial denaturation; 40 cycles of 10 s denaturation at 98 °C/20 s annealing at 50 °C/20 s elongation at 72 °C; 10 min at 72 °C final elongation 72 °C. The detailed standard reaction mix is provided in Appendix A. Fragments of relevant sizes were purified and cloned into the expression vector pQE30Xa (QIAGEN, Hilden, Germany). Successfully cloned cDNAs were sequenced for control purposes by Biosearch Technologies (LGC, Berlin, Germany).

### 4.5. Expression, Purification, Processing, and Quantification of Putative Decorsins

The detailed protocol to express, purify, process, and quantify the respective recombinant proteins was described in numerous recent publications (e.g., [26,54]). Briefly, all factors were expressed in the common laboratory strain *Escherichia coli* DH5α [69] at a cultivation temperature of 37 °C. After expression, cells were harvested, the pellet was resuspended in binding buffer (20 mmol/L Tris/HCl, 500 mmol/L NaCl, 5 mmol/L imidazole, pH 7.9), and the cells were sonicated using a Sonoplus homogenizer (Bandelin, Berlin, Germany). After centrifugation for 1 h at 4 °C and 4500 rpm (appr. 3900× *g*) in a Labofuge 400R (Thermo Scientific, Schwerte, Germany), the supernatant containing the cytosolic fraction was carefully removed and loaded onto a self-packed column containing Ni–NTA His-Bind^®^ resin (Merck, Darmstadt, Germany). Washing and elution steps were performed using a gradient of increasing imidazole concentrations (20 mmol/L Tris/HCl, 500 mmol/L NaCl, 5/20/50/100/200/500 mmol/L imidazole, pH 7.9) and a final stripping solution (20 mmol/L Tris/HCl, 500 mmol/L NaCl, 100 mmol/L Na_2_EDTA, pH 7.9). Equal volumes of every elution fraction were analyzed by SDS-PAGE on 20% gels. Prior to factor Xa protease treatment, fractions of interest were dialyzed two times for 24 h at 4 °C against a 100-fold excess of reaction buffer (20 mmol/L Tris/HCl, 50 mmol/L NaCl, 1 mmol/L CaCl_2_, pH 6.5). The pQE30Xa vector encodes a factor Xa protease recognition site that is located between the His-tag coding region at the 5′ side and the multiple cloning site at the 3′ side. A subsequent factor Xa protease treatment hence cleaves off the His-tag and results in a recombinant protein that lacks any vector-derived amino acid residues at the N-terminus. Factor Xa protease cleavage and removal were performed as recommended by the manufacturer (Qiagen, Hilden, Germany). Molar concentrations of final protein solutions were calculated by dividing the absorbance at 280 nm by the molar absorption coefficient according to the equation ε = (nW × 5500) + (nY × 1490) + (nC × 125) [70,71].

### 4.6. Platelet Aggregation Assays

All assays were performed with human blood samples that were obtained from healthy human volunteers after written informed consent and approval from the respective institutional ethics committee. The blood collection, the blood preparation, and the subsequent experimental procedure all followed the routine as described in Müller et al. [24]. Briefly, 10 mL of venous blood was taken from the antecubital vein of the human volunteer using an S-Monovette^®^ (Sarstedt, Nürnbrecht, Germany) prefilled with citrate buffer. The first centrifugation step was performed at 200× *g* for 20 min. After centrifugation, the supernatant (platelet-rich plasma, PRP) was transferred, and the remaining blood was centrifuged again for 10 min at 2000× *g*. The supernatant was dedicated as platelet-poor plasma (PPP), transferred, and used as a reference value for maximal platelet aggregation. For technical and logistical reasons, the measurements were performed using either a TA-8 V aggregometer (Diagnostica Stago S.A.S., Asnièressur-Seine, France) or an APACT-4004 aggregometer (LABiTec, Ahrensburg, Germany). The snake venom-derived platelet aggregation inhibitors tirofiban and eptifibatide (Sigma-Aldrich, Taufkirchen, Germany) were used as positive controls for complete inhibition of platelet aggregation. PRP was pre-incubated with the respective test and control compounds (final concentration 3.2 μmol/L) or buffer for 1 min at 37 °C. For the measurement, PRP was transferred into test cuvettes and stimulated with ADP (200 μmol/L; Hart Biologicals, Hartlepool, UK; final concentration 5 μmol/L) after 1 min of runtime. The final volume in each test cuvette was 250 μL (TA-8 V aggregometer) or 200 µL (APACT-4004 aggregometer) of diluted PRP. All experiments were performed at 37 °C over a time period of 400 s. Maximal aggregation in percentage and the area under the curve were calculated as quantitative output parameters [72]. Platelet aggregation tests were performed in two to four technical replicates.

## 5. Conclusions

Hematophagous leeches express a broad variety of bioactive compounds including numerous anticoagulants that affect both primary (platelet aggregation) and secondary (blood coagulation) hemostasis. The current study describes the identification and functional characterization of putative monomeric and oligomeric decorsins in the Asian medicinal leech *Hirudinaria manillensis*. Platelet aggregation inhibitors of the decorsin/ornatin-type have not been described in representatives of the genus *Hirudinaria* so far. Our investigations hence significantly expand the knowledge on the diversity of putative anticoagulants in leeches and provide a basis for further studies that may focus on the development of promising drugs for therapeutic applications. In addition, based on two key mechanisms of alternative pre-mRNA splicing, we outline a scenario that illustrates the amazing potential of just a single multimeric decorsin (and hirudin) gene to express an enormous variety of different (putatively) bioactive factors. Since each of these factors may display a specific biological effect, the “leech drug store” is probably even better filled than assumed to date.

## Figures and Tables

**Figure 1 ijms-26-11017-f001:**
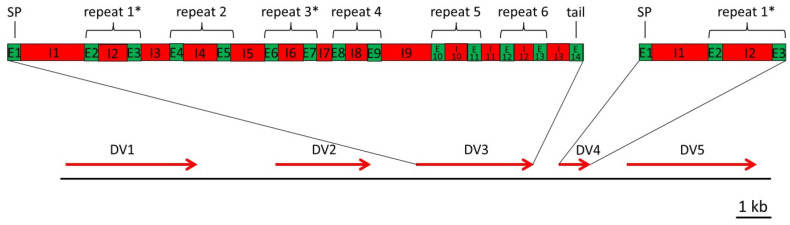
Schematic representation of a segment on genome contig00065 of *H. manillensis* comprising five putative decorsin genes (Hman_DV1-DV5). Red arrows indicate position, size, and orientation of the respective genes. The upper boxes illustrate the structures of two genes (left: putative decorsin gene Hman_DV3; right: putative decorsin gene Hman_DV4). Exons (E) are labelled in green, and introns (I) are labelled in red. The repeats of the two exons that encode the globular domain are marked with curved brackets. SP indicates the signal peptide; * indicates a repeat that contains an RGD motif; the bar indicates a length of 1 kb.

**Figure 2 ijms-26-11017-f002:**
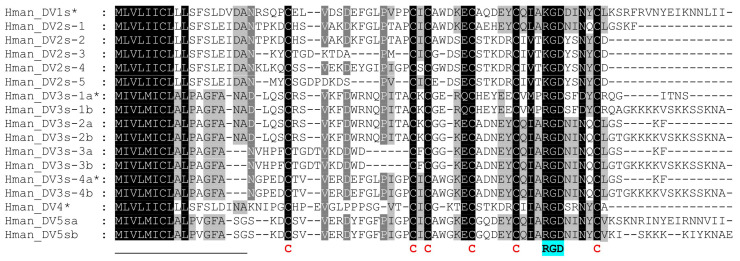
Multiple sequence alignment of putative monomeric decorsins of *H. manillensis*. Black background indicates conserved residues; grey background indicates similar residues. The six conserved cysteine residues that form the globular domain are marked in bold and red. The RGD/KGD motif is marked in bold and cyan. * indicates factors that were selected for functional analyses. Abbreviations are used according to the IUPAC code.

**Figure 3 ijms-26-11017-f003:**
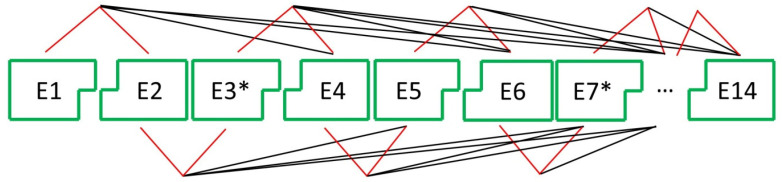
Schematic representation of a partial Hman_DV3 pre-mRNA of *H. manillensis*. E1, E2, …, E14 indicate the respective exons; * indicates the exons that contain a RGD motif. The red lines illustrate regular splicing; the black lines illustrate putative alternative splicing by exon skipping. Exon E1 encodes the signal peptide; exon E14 encodes the C-terminal tail. Exon-overlapping codons are illustrated by angled connections, and exon-non-overlapping codons are illustrated by blunt connections.

**Figure 4 ijms-26-11017-f004:**
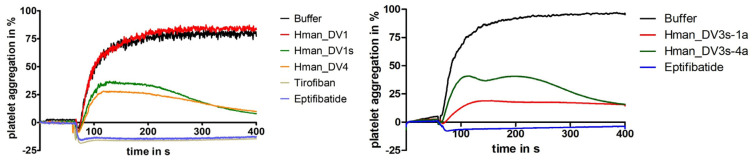
Standard platelet aggregation assays of decorsin variants Hman_DV1, Hman_DV1s, Hman_DV3s-1a, Hman_DV3s-4a, and Hman_DV4 of *H. manillensis*. Tirofiban and eptifibatide were used as positive control compounds for the complete inhibition of aggregation; buffer was used as a negative control. All test substances were used at a final concentration of 3.2 μmol/L. Platelet aggregation was induced by the addition of ADP to a final concentration of 5 μmol/L. Results are the mean of two to four independent measurements.

**Table 1 ijms-26-11017-t001:** Position and orientation (forward = fw or reverse and complementary = re+co) of putative decorsin genes and the hirudin/HLF/TH genes in the genomes of *H. manillensis*. In addition, the total lengths and the distances between the gene ranges are provided.

Gene	Guan et al., 2020 contig00065	Zheng et al., 2023 Chromosome 3	Liu et al., 2023 Chromosome 4
Hman_DV1	1,441,956–1,445,508 re+co	12,431,399–12,434,976 fw	12,102,877–12,106,430 fw
Hman_DV2	1,437,246–1,439,788 re+co	12,437,478–12,440,006 fw	12,108,608–12,111,161 fw
Hman_DV3	1,432,730–1,436,010 re+co	12,441,249–12,444,510 fw	12,112,407–12,115,657 fw
Hman_DV4	1,431,197–1,432,077 re+co	12,445,166–12,446,038 fw	12,116,319–12,117,203 fw
Hman_DV5	1,426,608–1,430,176 re+co	12,447,043–12,450,617 fw	12,118,197–12,120,025 fw
total length:	18,901 bp	19,219 bp	17,149 bp
gap:	140,046 bp	121,121 bp	133,389 bp
Hman_HV1	1,283,961–1,284,564 fw	12,587,203–12,587,806 re+co	12,249,296–12,249,899 re+co
Hman_HV2	1,287,886–1,288,539 fw	12,583,089–12,583,742 re+co	12,245,322–12,245,958 re+co
Hman_HV3	1,289,984–1,290,638 fw	12,580,970–12,581,635 re+co	12,243,197–12,243,865 re+co
Hman_HV4	1,297,158–1,297,813 fw	12,575,511–12,576,174 re+co	12,240,038–12,240,688 re+co
Hman_TH	1,300,423–1,301,910 fw	12,571,738–12,572,909 re+co	12,236,266–12,237,439 re+co
total length:	17,950 bp	16,069 bp	13,634 bp

**Table 2 ijms-26-11017-t002:** Lengths, number of repeats of the globular domain, sequence, and location (number of the respective repeat) of the RGD/KGD motifs and predicted molecular mass (MW) and isoelectric point (pI) values of putative decorsins of *H. manillensis*. * indicates factors that were selected for functional analyses.

Factor	Length in aa	Repeats	Motifs	MW in kDa	pI
Hman_DV1 *	257	6	1 (KGD)	28.81	5.49
Hman_DV2	171	4	1 (KGD) 4 (RGD)	19.12	4.58
Hman_DV3 *	271	6	1 (RGD) 3 (RGD)	30.69	8.26
Hman_DV4 *	45	1	1 (RGD)	4.75	8.33
Hman_DV5	268	6	1 (RGD)	30.17	8.60

**Table 3 ijms-26-11017-t003:** Lengths, predicted molecular mass (MW), and isoelectric point (pI) values of all putative monomeric decorsins of *H. manillensis* without the signal peptide sequences. * indicates factors that were selected for functional analyses.

Factor	Length in aa	MW in kDa	pI
Hman_DV1s *	63	7.29	4.50
Hman_DV2s-1	51	5.66	6.05
Hman_DV2s-2	47	5.19	5.53
Hman_DV2s-3	39	4.30	4.54
Hman_DV2s-4	47	5.23	4.72
Hman_DV2s-5	40	4.46	4.08
Hman_DV3s-1a *	51	5.99	5.66
Hman_DV3s-1b	61	7.09	9.17
Hman_DV3s-2a	49	5.52	7.76
Hman_DV3s-2b	61	6.75	9.33
Hman_DV3s-3a	46	5.12	4.40
Hman_DV3s-3b	58	6.36	8.28
Hman_DV3s-4a *	51	5.58	4.21
Hman_DV3s-4b	63	6.82	7.21
Hman_DV4 *	45	4.75	8.33
Hman_DV5sa	61	6.94	7.51
Hman_DV5sb	58	6.61	8.22

## Data Availability

The original contributions presented in this study are included in the article/Appendix A. Further inquiries can be directed to the corresponding author.

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
