# Peer review of "Monomeric and Oligomeric Decorsins of the Asian Medicinal Leech *Hirudinaria manillensis"

_ijms, 2025, doi:10.3390/ijms262211017_

Round 1
Reviewer 1 Report
Comments and Suggestions for Authors
Hematophagous leeches express a broad variety of bioactive compounds including numerous anticoagulants that affect both the primary (the platelet aggregation) and the secondary (the blood coagulation) hemostasis. The current study describes the identifica-tion and functional characterization of putative monomeric and oligomeric decorsins in the Asian medicinal leech Hirudinaria manillensis. These studies have greatly expanded our understanding of the diversity of potential anticoagulants in leeches and have provided a foundation for further research. Therefore, I suggest accepting it after minor revisions.
- "the Asian Medicinal Leech" should be placed before "Hirudinaria manillensis" in the title of the manuscript.
- The introduction section is quite lengthy, so it needs to be condensed.
- In line 387-389, the primers were designed based on "the respective genomic sequences". Please provide more specific parameters regarding the primer design (such as Tm value, expected product length), as well as details to ensure primer specificity to avoid non-specific amplification.
- In line 390-389, the manuscripts mentioned the use of a high-fidelity DNA polymerase, but did not provide a specific PCR reaction conditions (such as annealing temperature, extending time and cycle number, etc.). To ensure the reproducibility of the experiment, please provide a detailed PCR reaction system.
- In line 429, “Platelet aggregation tests were performed in two to four technical replicates” lacks statistical significance with only 2–4 replicates. Were biological replicates included?
- In Figure 4, there is a lack of statistical significance annotations..
- The contents of Table 1, Table 2 and Table 3 in the article are not arranged in a neat and consistent manner. Please make the necessary corrections.
- There are too many spaces in the references, so they need to be corrected.
The revised parts in the manuscript are for reference only.

Author Response
Comment 1: "the Asian Medicinal Leech" should be placed before "Hirudinaria manillensis" in the title of the manuscript.
Answer: We have changed the title of the manuscript according to the request of the reviewer.
Comment 2: The introduction section is quite lengthy, so it needs to be condensed.
Answer: We agree that the introduction is quite extensive, however, is was written with the intention to provide the relevant information to a (hopefully) broad audience with very different background knowledge. The manuscript addresses topics that might be of interest for readers that are more familiar with leech biology (chapters 2.1 and 2.3), but also for readers that are "hardcore" molecular biologists and may hence focus their attention on the alternative splicing model (chapter 2.2). Both putative groups of readers should be comprehensively informed. In addition, none of the two other reviewers complained about length and content of the introduction section, and we hence prefer to not shorten the text. We hope that the reviewer can follow our arguments.
Comment 3: In line 387-389, the primers were designed based on "the respective genomic sequences". Please provide more specific parameters regarding the primer design (such as Tm value, expected product length), as well as details to ensure primer specificity to avoid non-specific amplification.
Answer: The sequences of all primers that were used in the study are already provided in Supplementary Information Table S1, and we have now modified Table S1 and added both the Tm values of all primers and the expected amplicon lengths. As already stated in the manuscript, the primers were solely designed based on the respective genomic sequences that were used for the gene synthesis process to get the corresponding cDNAs in hand. The positions of the primers were hence fixed to allow the amplification of the desired fragments. Each pair of primers (fw and rev primers) were only used in a PCR reaction with the respective synthesized cDNA as the target, primer specificity was hence not an issue at all. All cloned amplicons were sequenced prior to further use.
Comment 4: In line 390-389, the manuscripts mentioned the use of a high-fidelity DNA polymerase, but did not provide a specific PCR reaction conditions (such as annealing temperature, extending time and cycle number, etc.). To ensure the reproducibility of the experiment, please provide a detailed PCR reaction system.
Answer: We have added the requested information on the PCR protocol to chapter 4.4. A detailed description of the PCR mix is provided in Supplementary Information Table S1.
Comment 5: In line 429, “Platelet aggregation tests were performed in two to four technical replicates” lacks statistical significance with only 2–4 replicates. Were biological replicates included?
Answer: It is common in clinical tests to run platelet aggregations assays with two technical replicates only. To run four replicates as it was done for the functional characterization of the putative decorsins is hence already above the requirements. Platelet aggregation assays are technically and logistically very challenging, and the main reason is the limitation that human platelets can only be obtained from a healthy voluntary donor and need to be freshly isolated and processed, but age quite rapidly and can only be used within a narrow time window of 2 - 3 hours as a maximum. It is hence good practice to conduct comparative aggregation measurements from the same platelet preparation and within a short time. Platelet preparations from different donors ("biological replicates") do not necessarily enhance the quality of the data.
Comment 6: In Figure 4, there is a lack of statistical significance annotations.
Answer: Platelet aggregation curves represent a qualitative marker and any changes in their shape and slope already indicate an interference with the platelet-aggregation process per se. Statistical significance analyses are hence uncommon. We are aware that the lack of a statistical analysis of the platelet-aggregation inhibition assay data can be considered as a limitation of our manuscript, but we are confident that the results of our investigations represent robust effects.
Comment 7: The contents of Table 1, Table 2 and Table 3 in the article are not arranged in a neat and consistent manner. Please make the necessary corrections.
Answer: We are very sorry for the misarrangement of the Tables. The submitted manuscript was correct, but was later formatted by the publisher to meet the journal style. We are in contact with the journal and will eventually fix the issue. For the moment we have integrated the Tables 1-3 as jpg images.
Comment 8: There are too many spaces in the references, so they need to be corrected.
Answer: The same problem: the submitted manuscript was fine, but the reference list was formatted by the publisher to meet the journal style. We hope to get that issue fixed, too.
Reviewer 2 Report
Comments and Suggestions for Authors
The article, titled "Monomeric and Oligomeric Decorsins of Hirudinaria 2 Manillensis, the Asian Medicinal Leech," meets all the criteria for being considered original research.
The experiment was carefully designed. The experimental procedure is described in detail in the "Materials and Methods" section, ensuring its reproducibility.
All data necessary for understanding the figures are presented not only in the figures themselves but also, crucially, in the legends.
The study is groundbreaking, as evidenced by the inclusion of recent literature in the discussion, including genomes published in 2020 and 2023.
It would be useful to include supporting results on the degree of purification of the obtained recombinant proteins. The method for obtaining and isolating proteins from the culture mass requires detailed description. Was the molecular weight of the obtained recombinant preparations determined? How was the conformity of the structure of the obtained recombinant proteins with the expected structure confirmed?
How can the negative platelet aggregation results for the positive controls be explained? For a correct assessment, their concentrations may have needed to be reduced.
The formatting of the tables needs to be corrected.
Author Response
Comment 1: The article, titled "Monomeric and Oligomeric Decorsins of Hirudinaria Manillensis, the Asian Medicinal Leech," meets all the criteria for being considered original research.
The experiment was carefully designed. The experimental procedure is described in detail in the "Materials and Methods" section, ensuring its reproducibility.
All data necessary for understanding the figures are presented not only in the figures themselves but also, crucially, in the legends.
The study is groundbreaking, as evidenced by the inclusion of recent literature in the discussion, including genomes published in 2020 and 2023.
Answer: We are very happy about the positive comment on our manuscript!
Comment 2: It would be useful to include supporting results on the degree of purification of the obtained recombinant proteins.
Answer: We have added gel images to illustrate the His-tag purification steps of all factors as Supplementary Information Figure S1. The images also provide information on the degrees of purity. Yield and degree of purity varied between the factors, but were in a reasonable range to proceed and to conduct the functional analyses. We are aware that none of the factors was "pure", the results of the platelet aggregation assays hence represent an underestimation of the real biological activities.
Comment 3: The method for obtaining and isolating proteins from the culture mass requires detailed description.
Answer: We have added a more detailed description of the expression and purification process of all factors to chapter 4.5.
Comment 4: Was the molecular weight of the obtained recombinant preparations determined?
Answer: All factors were analyzed by SDS-PAGE (Supplementary Information Table S1). We do not have the technical equipment and the expertise to perform more elaborate determinations.
Comment 5: How was the conformity of the structure of the obtained recombinant proteins with the expected structure confirmed?
Answer: The reviewer addresses a very good point. The correct structure of members of the hirudin superfamily critically depends on the correct formation of several disulfide bonds. A verification of the correct disulfide bond formation, however, can seriously be done only by crystallization and X-ray structure analysis of the respective factors, and this is far beyond our technical capabilities and methodological expertise. Native polyacrylamid gel analysis may provide some indications whether or not the respective recombinant protein exists in various conformations, but does not provide information which of these conformations is the correct one.
We are completely aware of the fact that the correct disulfide bond formation is crucial for the structure and consequently the function/activity of the recombinant decorsins, and we`ve addressed that important issue in a previous (Lukas et al. 2022, doi.org/10.1007/s00436-022-07634-0) and a very recent (Wang et al. 2025; doi.org/10.3390/mps8040089) publication. The clear platelet aggregation inhibitory effects of the monomeric decorsins indicate that at least a large fraction, if not the majority of the recombinant factors is correctly folded. However, the recombinant multimeric decorsin DV1 is very likely not correctly folded, and we have discussed that point in our manuscript.
Comment 6: How can the negative platelet aggregation results for the positive controls be explained? For a correct assessment, their concentrations may have needed to be reduced.
Answer: To explain the effect shown in the figure, we would like to explain the principle of LTA assays in more detail. Platelet aggregation is determined in “light transmission aggregometry” by measuring the light transmission of the platelet suspension. Non-aggregated platelets scatter more light. The solution appears cloudy. After adding an agonist such as ADP, the platelets, which are discoid in their non-activated state, initially undergo spherical deformation. This transition to a spherical state is referred to as “shape change.” This results in an increase in light scattering, which further reduces the passage of light through the platelet suspension. In the LTA assay, this manifests as a decrease in light transmission below the baseline. Eptifibatide and tirofiban, which are used here as “positive controls” for platelet aggregation inhibition, are potent inhibitors of GPIIb/IIIa fibrinogen receptors and completely prevent the platelet aggregation that usually occurs after shape change. Therefore, the reduction in light transmission remains below baseline as shown in the figure. The apparent “inverse” effect on platelet aggregation shown in the figure is thus an expression of the potent inhibition of GPIIb/IIIa receptors.
We have tested all putative decorsins of Hirudinaria manillensis at a final concentration of 3.2 µmol/l (see chapter 4.6). The value is based on measurements of a broad variety of putative decorsins in a couple of previous investigations. It turned out that this single value is very well suited to cover the whole range of different effects on platelet aggregation from "no inhibition of aggregation" to "eptifibatide/tirofiban-like effects". To allow a direct comparison within and between the measurements we also use the positive controls at a concentration of 3.2 µmol/l.
Comment 7: The formatting of the tables needs to be corrected.
Answer: We are very sorry for the misarrangement of the Tables. The submitted manuscript was correct, but was later formatted by the publisher to meet the journal style. We are in contact with the journal and will eventually fix the issue. For the moment we have integrated the Tables 1-3 as jpg images.
Reviewer 3 Report
Comments and Suggestions for Authors
This manuscript presents a systematic sequence analysis of the decorsin family in Hirudinaria manillensis, along with functional validation of several representative proteins. This work broadens our understanding of the antithrombotic protein family in Hirudinaria manillensis and is beneficial for the development of related antithrombotic drugs. The paper is rich in data, with sound analysis and reliable conclusions, making it suitable for publication. I have a few minor suggestions that I hope will help improve the quality of the manuscript.
(1) The manuscript involves complex gene family members and their sequences, which may hinder the reproducibility of the results for readers and other researchers. Although the authors have provided the coordinate ranges (Table 1) and putative monomeric sequences, the information is still not easily interpretable. It is recommended that the authors systematically organize the extracted sequences, providing the longest sequence for each gene with comprehensive annotations (using different colors to mark exons and introns), and submit this as supplementary material.
(2) The authors performed recombinant expression of several representative genes. It is requested that the codon-optimized sequences of these genes be provided in the supplementary materials to facilitate verification by other researchers.
(3) All tables lack borders and are not properly aligned. Please reformat them. For Table 3, it should be clarified whether the signal peptides were removed in the sequence analysis.
Author Response
Comment 1: This manuscript presents a systematic sequence analysis of the decorsin family in Hirudinaria manillensis, along with functional validation of several representative proteins. This work broadens our understanding of the antithrombotic protein family in Hirudinaria manillensis and is beneficial for the development of related antithrombotic drugs. The paper is rich in data, with sound analysis and reliable conclusions, making it suitable for publication.
Answer: We are very happy about the positive comment on our manuscript!
I have a few minor suggestions that I hope will help improve the quality of the manuscript.
Comment 2: The manuscript involves complex gene family members and their sequences, which may hinder the reproducibility of the results for readers and other researchers. Although the authors have provided the coordinate ranges (Table 1) and putative monomeric sequences, the information is still not easily interpretable. It is recommended that the authors systematically organize the extracted sequences, providing the longest sequence for each gene with comprehensive annotations (using different colors to mark exons and introns), and submit this as supplementary material.
Answer: We are a little confused about the remark since we have already entirely fulfilled the requirements of the reviewer. The detailed nucleotide sequence data for all five decorsin genes of H. manillensis are provided in the Supplementary Information files S2-S6 including full annotations of exons and introns using different colors, positions of the cysteine codons and the RGD/KGD motif codons and the derived amino acid sequences. All information that are needed for further analyses can be easily extracted from the files.
Comment 3: The authors performed recombinant expression of several representative genes. It is requested that the codon-optimized sequences of these genes be provided in the supplementary materials to facilitate verification by other researchers.
Answer: All synthesized genes/cDNAs contained the wildtype sequences, we did not order codon-optimized sequences. The sequence data that are presented in the Supplementary Information hence represent the sequences that were used in our experiments.
Comment 4: All tables lack borders and are not properly aligned. Please reformat them. For Table 3, it should be clarified whether the signal peptides were removed in the sequence analysis.
Answer: We are very sorry for the misarrangement of the Tables. The submitted manuscript was correct, but was later formatted by the publisher to meet the journal style. We are in contact with the journal and will eventually fix the issue. For the moment we have integrated the Tables 1-3 as jpg images.
The data that are presented in Table 3 are without the signal peptide sequences. We have added the missing information to the figure legend.
Round 2
Reviewer 2 Report
Comments and Suggestions for Authors
Accept in present form